# LLM360: Towards Fully Transparent Open-Source LLMs

**Zhengzhong Liu**[a][b]   **Aurick Qiao**[a]   **Willie Neiswanger**[c]   **Hongyi Wang**[d]   **Bowen Tan**[d]
**Tianhua Tao**[e]   **Junbo Li**[b]   **Yuqi Wang**[a]   **Suqi Sun**[a]   **Omkar Pangarkar**[a]   **Richard Fan**[a]
**Yi Gu**[f]   **Victor Miller**[a]   **Yonghao Zhuang**[d]   **Guowei He**[b]   **Haonan Li**[b]   **Fajri Koto**[b]
**Liping Tang**[b]   **Nikhil Ranjan**[b]   **Zhiqiang Shen**[b]   **Roberto Iriondo**[b]   **Cun Mu**[b]
**Zhiting Hu**[f]   **Mark Schulze**[a]   **Preslav Nakov**[b]   **Timothy Baldwin**[b]   **Eric P. Xing**[b][d]

[a]Petuum Inc.
*Pittsburgh, Pennsylvania, United States*
{hector.liu, aurick.qiao, yuqi.wang, suqi.sun, omkar.pangarkar
  richard.fan, victor.miller, mark.schulze}@petuum.com

[b]Mohamed bin Zayed University of Artificial Intelligence
*Abu Dhabi, United Arab Emirates*
{hector.liu, junbo.li, guowei.he, haonan.li, fajri.koto, liping.tang
  nikhil.ranjan, zhiqiang.shen, rob.iriondo, caris.mu, eric.xing}@mbzuai.ac.ae

[c]University of Southern California
*Los Angeles, California, United States*
{neiswang}@usc.edu

[d]Carnegie Mellon University
*Pittsburgh, Pennsylvania, United States*
{btan2, hongyiwa, yzhuang2, epxing}@andrew.cmu.edu

[e]University of Illinois Urbana-Champaign
*Champaign, Illinois, United States*
{tianhua3}@illinois.edu

[f]University of California San Diego
*La Jolla, California, United States*
{yig025, zhh019}@ucsd.edu

## Abstract

The recent surge in open-source Large Language Models (LLMs), such as LLaMA, Falcon, and Mistral, provides diverse options for AI practitioners and researchers. However, most LLMs have only released partial artifacts, such as the final model weights or inference code, and technical reports increasingly limit their scope to high-level design choices and surface statistics. These choices hinder progress in the field by degrading transparency into the training of LLMs and forcing teams to rediscover many details in the training process. We present **LLM360**, an initiative to fully open-source LLMs, which advocates for all training code and data, model checkpoints, and intermediate results to be made available to the community. The goal of LLM360 is to support open and collaborative AI research by making the end-to-end LLM training process transparent and reproducible by everyone. As a first step of LLM360, we release two 7B parameter LLMs pre-trained from scratch, AMBER and CRYSTAL, including their training code, data, intermediate checkpoints, and analyses (at llm360.ai). We are committed to continually pushing the boundaries of LLMs through this open-source effort. More large-scale and stronger models are underway and will be released in the future.

# 1 Introduction

The landscape of Large Language Models (LLMs) has experienced a remarkable transformation in the past one year, witnessing an unprecedented surge in both the popularity and capabilities of these models. At the forefront of this evolution are proprietary LLMs such as GPT-4 (OpenAI, 2023) and Claude (Claude, 2023), which have captured the attention of the AI community due to their power and versatility. At the same time, the recent emergence of openly accessible yet highly capable LLMs such as LLaMA (Touvron et al., 2023a;b), Falcon (Penedo et al., 2023), and Mistral (Jiang et al., 2023) have allowed researchers and practitioners at large to easily obtain, customize, and deploy LLMs in more diverse environments and for more diverse use cases.

Despite the growing influence of open-source LLMs, a notable trend has been to restrict visibility and access to their training, fine-tuning, and evaluation processes, including crucial components such as their training code and data. This practice limits the ability of the broader AI research community to study, replicate, and innovate upon advanced LLMs. A more transparent approach to sharing not just the final model but also training details and artifacts is crucial for fostering a more inclusive and collaborative research environment. Motivated by the above, we note the following specific challenges in LLM research today.

**Data Provenance.** Understanding the origins and characteristics of training data is crucial for assessing the reliability and biases inherent in LLMs. A lack of transparency about data sources and composition hinders the ability to identify and mitigate biases which can be perpetuated in model outputs. Simultaneously, data leakage—where training datasets overlap with benchmark datasets—can lead to misleading performance metrics that obscure a model's general effectiveness (studied in Wei et al. (2023); Zhou et al. (2023)). These issues highlight the need for clear documentation of data origins and usage in LLM development.

**Reproducibility.** Even with full disclosure of data sources, the lack of access to complete training code, configuration details, and specific datasets can make it challenging to reproduce the results reported in studies. For example, although training data mixtures are disclosed by LLaMA (Touvron et al., 2023a), the data processing and training code are not released. Yet, LLMs trained using an open reproduction of LLaMA's data (*e.g.,* RedPajama (Together Computer, 2023b;a)) do not fully reproduce its benchmark evaluations (Geng & Liu, 2023), indicating that additional data processing or training procedures may be necessary.

**Open Collaboration.** The practice of only releasing final model weights not only leads to redundant efforts but also poses unique challenges in conducting certain research. For instance, research into the emergent abilities of LLMs (Biderman et al., 2023a; Wei et al., 2022) or the investigation of how different training data affects model behavior (Yu et al., 2023; Xie et al., 2023) becomes more challenging without access to intermediate training checkpoints. Researchers are often forced to either work with the final model, which offers limited insights into its developmental nuances, or start from scratch, leading to unnecessary duplication of work and expenditure of compute.

LLM360[1] aims to address the issues above through a comprehensive open-source LLM effort. Models in LLM360 are published with all training and model details (*e.g.,* hyperparameters, schedules, architecture, and designs), all intermediate model checkpoints saved during training, and full disclosure of the exact pre-training data used.

Our contributions are:

- We outline the LLM360 approach, focusing on its design principles and the rationale for fully open-sourcing LLMs. We detail the core open-sourced artifacts, including datasets, code and configurations, model checkpoints, and metrics. We provide a target for transparency that all present and future LLM360 models strive to meet.

- We pretrain two new LLMs from scratch and release them under the LLM360 framework. The first LLMs in the LLM360 series: AMBER is a 7B English LLM pretrained on 1.3T tokens. CRYSTAL is a 7B English and code LLM pretrained on 1.4T tokens. We discuss

---

[1]The name LLM360 signifies open-sourcing LLMs from all angles, and that 360 data points (*i.e.,* checkpoints, data chunks, evaluation results) are released for our first model.

the development details, preliminary evaluations, observations, and lessons we learned from AMBER and CRYSTAL.

- We release all training code, pretraining data, model checkpoints, and evaluation metrics collected during pretraining for both models. Notably, AMBER is released with 360 model checkpoints saved during training, and CRYSTAL with 152.

LLM360's goal is to continuously open-source LLMs at various scales and on vaious domains, to fill the gap in the open source and research community.

The rest of this paper is organized as follows. In §2, we discuss related work and the predecessors that inspired LLM360. In §3, we provide a description of the LLM360 framework and the release artifacts that fall into its purview. In §4, we discuss the AMBER and CRYSTAL models, and preliminary analyses of both. §7 concludes.

## 2  Related Work

The closest previous work to LLM360 is Pythia, which ensures full reproducibility (Biderman et al., 2023b). The Pythia project provided 154 checkpoints for model sizes from 70M to 12B with the same data order to better support research on interpretability, learning dynamics, ethics and transparency. While Pythia is a pioneering work, it no longer reflects many recent LLM practices, such as training with more tokens (Touvron et al., 2023a) than the compute-optimal size (Hoffmann et al., 2022), or multi-phased, curriculum training on different data mixes (Gemini-Team et al., 2023; InternLM Team, 2023). On the other hand, LLM360 takes an approach prioritizing transparency and reproducibility under which up-to-date models can continue to be released, and our new 7B models surpasses the 12B Pythia model in public benchmarks (Beeching et al., 2023). Overall, Pythia set an early precedent for transparency and reproducibility of LLMs that we aim to perpetuate and expand in LLM360 to modern LLM pretraining regimes.

| LLM Name | Release Date | Pretraining | | Checkpoints | | Pretraining Dataset | | | Tokens |
|---|---|---|---|---|---|---|---|---|---|
| | | Code | Config | Model | Optim | Data Mix | Ordering | Available | ($T$) |
| GPT-J (Wang & Komatsuzaki, 2021) | May'21 | ✓ | ✓ | ✓ | ✓ | ✓ | ✓ | ✓ | 0.40 |
| GPT-NeoX (Black et al., 2022) | Apr'22 | ✓ | ✓ | ✓ | ✓ | ✓ | ✓ | ✓ | 0.40 |
| OPT (Zhang et al., 2022) | May'22 | ✓ | ✓ | ✓ | | ✓ | | | 0.18 |
| BLOOM (Scao et al., 2022) | Nov'22 | ✓ | ✓ | ✓ | ✓ | ✓ | ✓ | ✓ | 0.34 |
| Pythia (Biderman et al., 2023b) | Feb'23 | ✓ | ✓ | ✓ | ✓ | ✓ | ✓ | ✓ | 0.30 |
| LLaMA (Touvron et al., 2023a) | Feb'23 | | ✓ | | | ✓ | | | 1.0 |
| OpenLLaMA (Geng & Liu, 2023) | May'23 | ✓ | ✓ | ✓ | | ✓ | | ✓ | 1.0 |
| INCITE (Together Computer, 2023a) | May'23 | ✓ | ✓ | ✓ | | ✓ | | ✓ | 1.0 |
| MPT (MosaicML NLP Team, 2023) | May'23 | ✓ | ✓ | | | ✓ | | | 1.0 |
| Falcon (Almazrouei et al., 2023) | May'23 | | ✓ | | | ✓ | | | 1.5 |
| Llama 2 (Touvron et al., 2023b) | Jul'23 | | ✓ | | | | | | 2.0 |
| Qwen (Bai et al., 2023) | Aug'23 | | ✓ | | | | | | 2.4 |
| Mistral (Jiang et al., 2023) | Sep'23 | | | | | | | | ? |
| Yi (01.ai, 2023) | Nov'23 | | | | | | | | ? |
| AMBER | Dec'23 | ✓ | ✓ | ✓ | * | ✓ | ✓ | ✓ | 1.3 |
| CRYSTAL | Dec'23 | ✓ | ✓ | ✓ | ✓ | ✓ | ✓ | ✓ | 1.4 |
| OLMo 7B[†] (Groeneveld et al., 2024) | Feb'24 | ✓ | ✓ | ✓ | ✓ | ✓ | ✓ | ✓ | 2.46 |

Table 1: Summary of notable open-source LLMs. We note a trend of progressively less disclosure of important pretraining details over time: (1) availability of pretraining code, (2) disclosure of training configurations and hyperparameters, (3) intermediate checkpoints of model weights, (4) intermediate checkpoints of optimizer states, (5) disclosure of data mixture and sources, (6) reproducibility of pretraining data sequence, and (7) availability (or reconstruction scripts) of the pretraining data.
* *Amber optimizer states are lost due to errors in early implementation.*
[†] *We show the configuration of OLMo 7B; other OLMo models release similar artifacts.*

In general, open-source LLMs span a wide spectrum of transparency and reproducibility when it comes to their release artifacts. Many recent LLMs only release their final model architecture and weights, keeping their data sources and most training details

undisclosed (Touvron et al., 2023b; Bai et al., 2023; Jiang et al., 2023; 01.ai, 2023). Some are trained on publicly available datasets (Wang & Komatsuzaki, 2021; Black et al., 2022; Scao et al., 2022; Biderman et al., 2023b; Geng & Liu, 2023; Together Computer, 2023a; Shen et al., 2023), whereas others disclosed their data mixtures but do not make training-ready data available to the public (Zhang et al., 2022; Touvron et al., 2023a; MosaicML NLP Team, 2023; Almazrouei et al., 2023). Several LLMs of note have been released with substantially more transparent details and artifacts. The release of GPT-J (Wang & Komatsuzaki, 2021) and GPT-NeoX (Andonian et al., 2023) included training code, datasets, and up to 150 intermediate model checkpoints. The value of the open-source GPT-NeoX training code was demonstrated by its use in subsequent LLM pretraining by others in the community (Together Computer, 2023a; MosaicML NLP Team, 2023). The Stack datasets from the StarCoder series (Li et al., 2023; Lozhkov et al., 2024) has significantly facilitated the development of effective coding models. INCITE (Together Computer, 2023a), MPT (MosaicML NLP Team, 2023), and OpenLLaMA (Geng & Liu, 2023) were released with training code and training dataset, with INCITE also releasing 10 intermediate model checkpoints. After the LLM360 release, we are thrilled to see the community is joined with more open sourced LLMs (Groeneveld et al., 2024; Zhang et al., 2024; Üstün et al., 2024) similar to the LLM360 setting.

Overall, we observe a trend that (before late 2023), more recent and capable LLMs are becoming more closed in their release artifacts. In contrast, the goal of LLM360 is to release modern and high-quality models while maintaining a high degree of release transparency.

## 3 The LLM360 Initiative

In this section we present LLM360, an initiative to open source LLMs that promotes transparency, reproducibility, data/model provenance, and collaborative research. LLM360 provides guidance and recommendations for release artifacts that are collected during LLM pre-training and subsequently made publicly available to the community.

As part of the launch of LLM360, we also release two new pre-trained LLMs, which we hope will foster immediate interest and collaboration in the open-source research community. First, AMBER, an English language LLM with 6.7B parameters trained on 1.25 trillion tokens. Second, CRYSTAL, an English and code LLM, also with 6.7B parameters, trained on 1.4 trillion tokens. Details on AMBER and CRYSTAL are reported in §4.

**Training Dataset and Data Processing Code.** The pretraining dataset is a main ingredient of an LLM and it is important for users and adopters to have visibility into this data to assess potential behavior issues and biases. For example, recent concerns about benchmark data leakage into LLM pretraining is much easier to study when pretraining datasets are available (Zhou et al., 2023; Wei et al., 2023).

Furthermore, visible data improves the extensibility of LLMs in further training. For example, recent work suggests that training on repeated data disproportionately degrades final model performance (Hernandez et al., 2022). Given the breadth of data in modern pretraining, visibility into the original pretraining data is essential for avoiding repeated data in downstream fine-tuning or continual pretraining on specialized domains.

LLM360 advocates for the public release of the pre-training data. When applicable, details about data filtering, processing, and training order should be released as well. Doing so equips the community with better tools to assess the capabilities and risks of LLMs and to reproduce and build upon existing LLMs for future use cases.

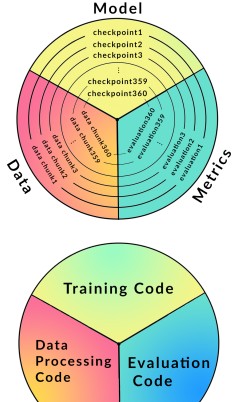

Figure 1: Artifacts released in LLM360 include data chunks, model checkpoints, and evaluation metrics, for up to 360 time stamps of training (and code for all parts).

**Training Code, Hyperparameters, and Configurations.** These code and settings have a significant impact on the performance and quality of LLM training, and are not always publicly disclosed. For example, we observed that a carefully balanced hybrid data-model-pipeline (3D) parallelism (Narayanan et al., 2021) can outperform the standard FSDP in PyTorch by up to

15% on our Nvidia A100 clusters. Another example is that it is essential to keep the inverse frequency matrix in RoPE positional embedding in FP32 (Su et al., 2021), which aligns with the observation in Qwen (Bai et al., 2023).

In LLM360, we open-source all of our pre-training frameworks, hyperparameters, and configurations, including the full training source code, training parameters such as learning rates and batch sizes, and system configurations such as parallelism dimensions.

**Model Checkpoints.**    It is typical during LLM training to periodically save checkpoints of the model to persistent storage. These checkpoints are not only crucial for recovery from faults during training, but also useful in post-training research such as studying different data and/or hyperparameter schedules, or reproducing infrequently-occurring training faults (*e.g.,* loss spikes, NaN results). Recent research on model quantization and compression heavily relies on analysis of model weights and the dynamics during training (Dettmers et al., 2022; Liu et al., 2023).

LLM360 models are published with all intermediate checkpoints saved during their training, including model weights and optimizer states (when applicable, *e.g.,* Adam (Kingma & Ba, 2017) moving averages). These checkpoints enable continued training from a range of starting points without training from scratch, making it easier to study and reproduce a wider variety of effects during training.

**Metrics.**    LLMs undergo training over weeks to months, and the trends and evolution patterns over this training period can offer valuable information. However, access to detailed logs and intermediate metrics for LLMs is typically limited. These statistics often contain key insights that cannot be directly derived, and even a simple analysis on the metrics, such as computing metric variances or norms, can reveal significant findings. For instance, the team behind GLM proposed an effective gradient shrinking algorithm for handling loss spikes and NaN losses by analyzing gradient norm behaviors (Zeng et al., 2023).

Our aim with LLM360 is to alleviate this problem by completely open sourcing the logs and metrics we collect. This includes system statistics (e.g., GPU workload), training logs (e.g., loss, gradient norm), and evaluation metrics (e.g., perplexity, downstream tasks). Access to these logs may facilitate a deeper understanding of the whole training process, including how LLMs evolve during various training scenarios. We provide easy access to the figures by sharing directly on the LLM360 Weights & Biases page[2]. A few example metrics include downstream evaluation results, training loss, gradient norm, etc.

In §5.1, we introduce how one can make use of the metrics, and illustrate an experiment tracking the memorization behavior of a model throughout training. The metrics are released in coordination with the data chunks and checkpoints for researchers to easily find their correspondence. Furthermore, we provide open access to the analysis and evaluation code used to foster reproducibility.

## 4    Amber and Crystal Pre-trained LLMs

In this section, we introduce two model families: AMBER, the first model in the LLM360 family, along with finetuned models AMBERCHAT and AMBERSAFE; and CRYSTAL, a code-focused model, along with the finetuned model CRYSTALCHAT.

Below we review the details of our pre-training dataset, including data preprocessing, format, data mixing ratios, architectural details, and specific pre-training hyperparameters. We also publish all details in the LLM360 code base.

**Model Configurations.**    We summarize all details in Table 2. AMBER uses a similar model architecture as LLaMA 7B, including model dimensions, use of RMSNorm (Zhang & Sennrich, 2019b), and rotary positional embeddings (RoPE) at each layer of the network (Su et al., 2021). CRYSTAL further incorporates maximal update parameterization ($\mu P$) (Yang et al., 2022), restricts RoPE to the first 25% of hidden dimensions, uses LayerNorm instead of RMSNorm, and makes minor adjustments to vocabulary size and intermediate hidden dimensions.

---

[2]`https://wandb.ai/llm360`

| AMBER | | CRYSTAL | | |
|---|---|---|---|---|
| Subset | Tokens | Phase | Subset | Tokens |
| Arxiv | 30B | 1 | SlimPajama (50%) | 345B |
| Book | 29B | 2 | SlimPajama (50%) | 927B |
| C4 | 198B | | StarCoder (×2) | |
| Refined-Web | 665B | | StarCoder (Python) | |
| StarCoder | 292B | 3 | StarCoder (JS) | 100B |
| StackExchange | 22B | | StarCoder (HTML) | |
| Wikipedia | 24B | | StarCoder (CSS) | |
| Total | 1.26T | Total | | 1.38T |

Figure 2: Data mixture in AMBER and CRYSTAL.

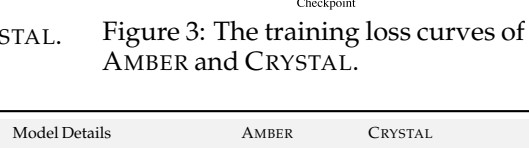

Figure 3: The training loss curves of AMBER and CRYSTAL.

**Pre-training Data.** Summarized in Figure 2, AMBER uses a pre-training dataset similar to OpenLLaMA (Geng & Liu, 2023)[3], which is sourced from a mixture of Refined-Web (Penedo et al., 2023), StarCoder (Li et al., 2023), RedPajama-v1 (Together Computer, 2023b), and C4 (Raffel et al., 2023), totaling 1.26 trillion tokens.

CRYSTAL uses a blend of SlimPajama (Soboleva et al., 2023) and StarCoder data (Li et al., 2023) with 1.38 trillion tokens in total. CRYSTAL's training process is divided into three stages. The first stage uses half of the SlimPajama data. The second stage uses the remaining half of the SlimPajama data, mixed with two epochs of StarCoder data. Finally, the third stage uses Python and several web-related subsets from StarCoder, mixed with a small subset of 10 billion tokens from SlimPajama. All preprocessed data and data mixing scripts are released in LLM360 Hugging Face and Github repositories.

| Model Details | AMBER | CRYSTAL |
|---|---|---|
| Number Parameters | 6.7B | 6.7B |
| Hidden Size | 4096 | 4096 |
| Intermediate Size (in MLPs) | 11008 | 10922 |
| Number of Attention Heads | 32 | 32 |
| Number of Hidden Layers | 32 | 32 |
| LR Schedule | Cosine Decay | Linear Decay |
| Normalization | RMSNorm | LayerNorm |
| Activation | SwiGLU | SwiGLU |
| Sequence Length | 2048 | 2048 |
| Vocabulary Size | 32000 | 32032 |
| Position Embedding | Rotary | Rotary (25%) |
| Bias | None | Linear & LayerNorm |
| QK Dot Product Scaling | $\frac{QK^T}{\sqrt{d}}$ | $\frac{QK^T}{d}$ |
| Model Parametrization | Standard | $\mu P$ (Yang et al., 2022) |
| Warmup Steps | 2000 | multi-phase |
| Batch Size | 2240 | 2112 |

Table 2: Model architecture comparison. Several choices are hardware dependent. See §B for the full list of hyperparameters and details.

**Pre-training Infrastructure.** AMBER is trained on a GPU cluster of 56 DGX A100 nodes, each equipped with 4× 80GB A100 GPUs. Additional details on the cluster infrastructure can be found in Appendix C. CRYSTAL is trained on the Cerebras Condor Galaxy 1 (CG-1), a 4 exaFLOPS, 54 million core, 64-node cloud AI supercomputer[4].

### 4.1 Finetuned Models

The instruction following ability (Ouyang et al., 2022) of language models has significantly simplify the adoption of LLMs in pratical applications. Similar methods are adopted by the open source community (Taori et al., 2023; Chiang et al., 2023). We released both supervised instruction finetuning and preference tuning versions of the models.

**AMBERCHAT and AMBERSAFE.** We train AMBERCHAT, a supervised finetuning model, on enhanced ShareGPT data (Xu et al., 2023). We use FastChat (Zheng et al., 2023) to finetune the model for 3 epochs, where learning rate is $2 \times 10^{-5}$, gradient accumulation steps is 16, and warmup ratio is 0.04. We further create a preference tuned version AMBERSAFE, using Direct Parameter Optimization (DPO) (Rafailov et al., 2023), using the SafeRLHF dataset (Ji et al., 2023). Here we set $\beta$ to 0.1, gradient accumulation steps to 4, and the learning rate to $5 \times 10^{-7}$.

---

[3]https://github.com/openlm-research/open_llama#dataset-and-training
[4]https://www.cerebras.net/condor-galaxy-1

**CRYSTALCHAT.** The selection of finetuning data for CRYSTALCHAT features both language and coding proficiency. We use a 1.1 billion token dataset collection (see Appendix G.2 for details). CRYSTALCHAT is proficient in programming and excels in web programming.

The goal of AMBERCHAT and CRYSTALCHAT are to produce instruction-following models which are commonly used for research purposes. An additional goal of AMBERSAFE is to carry out a stage of safety-alignment in addition to instructing-tuning, so that the effects of each stage can be separately studied, whereas most open-weight models today only release the final result after both stages.

## 5 Amber and Crystal Evaluation

**Benchmark Results.** We conduct a wide range of evaluation on our models, and release the evaluation results on checkpoints throughout the training for observation across the pretraining process. All metrics are publicly available[5].

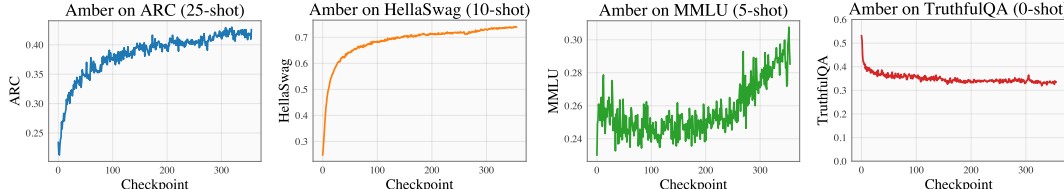

Figure 4: Results for AMBER on the Open LLM leaderboard metrics.

Figure 4 shows four benchmark metrics for AMBER: ARC (Clark et al., 2018), HellaSwag (Zellers et al., 2019), MMLU (Hendrycks et al., 2021), and TruthfulQA (Lin et al., 2022), evaluated with the Open LLM Leaderboard[6] settings. We can see that the HellaSwag and ARC evaluation scores monotonically increase during pre-training, while the TruthfulQA score decrease as the training proceeds. Interestingly, the MMLU score decreases and fluctuates in the initial stage of pretraining and then starts to increase at around 70% progress.

Figure 5 shows the metrics for CRYSTAL, with two additional metrics evaluating code generation: MBPP (Austin et al., 2021) and HumanEval (Chen et al., 2021). For most of the language metrics, we oberve a drop during transitions between each stage, since there is a drastic shift of domain composition (Table 2). Interestingly, the performance on MMLU consistently improves over phases 1 and 2, with scores starting to rise above a random guess quickly in stage 2, in contrast to the trend observed in AMBER.

In Table 3, we compare the final model performance of AMBER to a set of models trained around a similar time, namely OpenLLaMA, RedPajama-INCITE, Falcon, MPT. Many are inspired by the design of LLaMA. We find that AMBER is relatively competitive in scores such as MMLU, but its performance on ARC is behind the curve. We also find that our fine-tuned AMBER models are relatively strong, even compared with other similar models. In our early study, we note that AMBERCHAT simply trained on ShareGPT data also demonstrates much higher performance than our base model, which is slightly different from the trends shown on other models in the table. We leave further investigation of this to future work. As expected, AMBERSAFE significantly boosts the TruthfulQA score after aligning with safety preferences.

| The LLMs | ARC-C | HellaSwag | MMLU | TruthfulQA | Avg. |
|---|---|---|---|---|---|
| LLaMA2-7B-chat | 52.9 | 78.55 | 48.32 | 45.57 | 56.34 |
| LLaMA2-7B | 53.07 | 77.74 | 43.8 | 38.98 | 53.39 |
| AMBERSAFE (7B) | 45.22 | 74.14 | 37.78 | 55.44 | 53.15 |
| LLaMA-7B | 50.94 | 77.8 | 35.67 | 34.34 | 49.69 |
| AMBERCHAT (7B) | 42.83 | 74.03 | 38.88 | 40.72 | 49.12 |
| OpenLLaMA-v2-7B | 43.69 | 72.2 | 41.29 | 35.54 | 48.18 |
| MPT-7B | 47.7 | 77.57 | 30.8 | 33.44 | 47.38 |
| Falcon-7B | 47.87 | 78.13 | 27.79 | 34.26 | 47.01 |
| RedPajama-INCITE-7B-Instruct | 44.11 | 72.02 | 37.61 | 33.96 | 46.93 |
| Falcon-7B-instruct | 46.16 | 70.85 | 25.66 | 44.07 | 46.69 |
| OpenLLaMA-v1-7B | 47.01 | 71.98 | 30.49 | 34.85 | 46.08 |
| AMBER (7B) | 41.89 | 74.14 | 30.76 | 34.00 | 45.20 |
| RedPajama-INCITE-7B-Base | 46.25 | 71.63 | 27.68 | 33.03 | 44.65 |

Table 3: Open LLM leaderboard metrics comparison.

---

[5] https://wandb.ai/llm360
[6] https://huggingface.co/spaces/HuggingFaceH4/open_llm_leaderboard

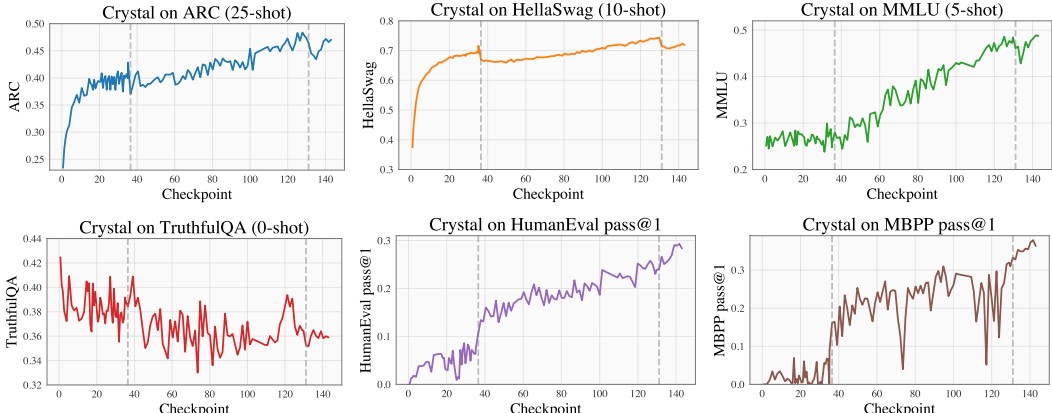

Figure 5: Results for CRYSTAL on the Open LLM leaderboard metrics. Grey vertical dashed lines denote the transition between the three stages of training.

Table 4 shows the comparison of CRYSTAL with popular open source models, including Falcon, OpenLLaMA-v2, LLaMA (1, 2), CodeLlama, Mixtral, and Starcoder. The performance of CRYSTAL is competitive among the models while maintaining a good balance between language and coding tasks. We see a high training efficiency by comparing the model trained with similar amount of tokens, achieving 48.78 on MMLU trained on only 1.27T tokens, surpassing LLaMA2-7B's 43.80.

The performance of AMBER model lags slightly behind some, especially newer 7B models. While our results are close to models released around the same time. We hypothesize this is due to the dataset choices, and the use of an incorrect precision when storing early checkpoints (Appendix E). Our next training effort CRYSTAL, with a more careful data strategy (*e.g.*, curriculum, deduplication), produced significantly better results on benchmarks. To evaluate our tuned models, we evaluated AMBER, AMBERCHAT, and AMBERSAFE with MT-Bench (Zheng et al., 2023). The models scored 2.49, 4.95, and 4.97, respectively (Table 9).

## 5.1 ANALYSIS360

Prior work such as Pythia (Biderman et al., 2023b) has shown insight results by analyzing intermediate model checkpoints during training. Our hope is that LLM360 can also provide the community with useful resources for both reference and research purposes. To this end, we release the initial version of the ANALYSIS360 project, an organized repository to analyze various aspects of the model behavior, including model characteristics and downstream evaluation results.

As an example of the analysis that can be performed using the full set of model checkpoints, we conduct an initial study on memorization in LLMs. Recent work (Carlini et al., 2021; 2022) shows that LLMs may memorize a significant part of their training data, which can be extracted with appropriate prompting. Such memorization not only raises privacy concerns in leaking private training data, but also downgrades the performance of LLMs if the training data contains unintended duplicates or peculiarities. To study this behavior, we carry out memorization experiments using the set of checkpoints and data from AMBER.

We adopt the *memorization score* introduced in (Biderman et al., 2023a) and also test 32-extractable (Carlini et al., 2021) sequences (see § D for full details and definitions). We sample 1000 sequences from each of the 360 data chunks, and use the first 64 tokens of each sequence to conduct experiments. Results are shown in Figure 6. On the left we show the distribution of memorization scores over 10 selected checkpoints. On the right, we group the data chunks based on their next selected checkpoint and plot the memorization score for each data chunk for each checkpoint. As demonstrated, the LLM360 released artifacts enable reproducible research and provide means for the community to study LLM technology.

## 6 Discussion and Take-home Messages

Pre-training LLMs is a computationally daunting task that many academic labs or small organizations cannot afford. We hope LLM360 provides comprehensive knowledge, allowing

| The LLMs | # Tokens | Language Tasks | | | | | Coding Tasks | | | Avg. |
|---|---|---|---|---|---|---|---|---|---|---|
| | | ARC-C | HellaSwag | MMLU | TruthfulQA | Avg. | HumanEval | MBPP | Avg. | |
| Mistral-7B | - | 59.98 | 83.31 | 64.16 | 42.15 | 63.40 | 29.12 | 38.78 | 33.95 | 48.68 |
| CRYSTAL (7B) | 1.38T | 47.01 | 71.97 | 48.78 | 35.91 | 50.92 | 28.38 | 36.38 | 32.38 | 41.65 |
| CodeLlama-7B | 2.5T | 39.93 | 60.80 | 31.12 | 37.82 | 42.42 | 33.50 | 41.40 | 37.45 | 39.94 |
| OpenLLaMA-v2-7B | 1T | 43.69 | 72.20 | 41.29 | 35.54 | 48.18 | 15.32 | 12.69 | 28.01 | 38.10 |
| LLaMA2-7B | 2T | 53.07 | 77.74 | 43.80 | 38.98 | 53.39 | 13.05 | 20.09 | 16.57 | 34.98 |
| LLaMA-7B | 1.4T | 50.94 | 77.80 | 35.67 | 34.34 | 49.69 | 10.61 | 17.04 | 13.83 | 31.76 |
| Falcon-7B | 1.5T | 47.87 | 78.13 | 27.79 | 34.26 | 47.01 | 9.42 | 13.39 | 11.41 | 29.21 |
| StarCoder-15B | 1.03T | – | – | – | – | – | 33.63 | 43.28 | 38.46 | – |

Table 4: Evaluation comparisons among a few notable code and language base models. The last column is the average of the language task average and the code task average. CRYSTAL strikes a good balance between both language and code tasks.

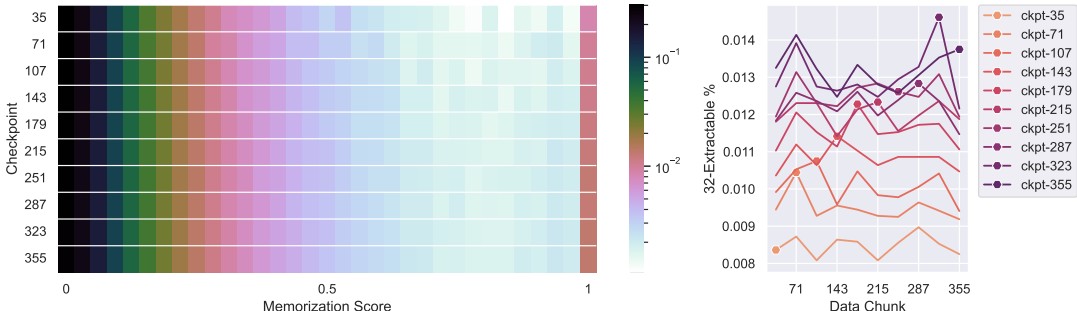

Figure 6: Left: Each row shows a histogram of memorization scores for a checkpoint. Similar to Biderman et al. (2023a), we observe a spike at score = 1, indicating that there are a larger number of memorized tokens than the threshold (32) used in the experiment. Right: Each curve shows the memorization scores over the data chunks, The marked spots indicate the latest chunk seen by that checkpoint; the region to the right of each mark indicates unseen data. We see that each checkpoint memorizes more of the latest seen data. The score drops after first seen data, but can increase with additional training.

readers to understand the details of pre-training (*e.g.*, loss curve behaviors, how evaluation metrics emerge) without needing to do so themselves. We also provide some of our learnings and potential use cases showing how one can use LLM360 for their own projects.

**Take-home Messages**

- In AMBER, NaN losses were periodically observed, which may have been caused by certain random states, training precision, or data quality issues. Our solutions included switching to a different random seed or skipping those data chunks. We noticed some "misbehaved" data chunks can cause NaN loss regardless of when they appear in training—for example, even after we moved them to the end of the training.

- We observed that a hybrid and carefully tuned parallelism strategy—combining data, tensor-model, and pipeline (also referred to as 3D) parallelism strategies (Narayanan et al., 2021)—achieves better system throughput than FSDP (Zhao et al., 2023), especially in distributed clusters with limited inter-node bandwidth.

- Data quality, mixing, and sequencing are crucial aspects of LLM pre-training. For AMBER, we adhered as closely as possible to the settings in LLaMA. However, AMBER's performance still lags behind LLaMA's—a key omission in LLaMA's technical report is a detailed description of its data preparation. On the other hand, our CRYSTAL dataset and which mixes English and code in three stages, achieves competitive performance with LLaMA on both the Open LLM Leaderboard and Code Evaluation benchmarks.

**Potential Use Cases of LLM360**

- One can conduct experimental studies at any stage of model training. As previously mentioned, the optimal data mixing ratio remains a significant open problem in LLM pre-training. However, it is nearly impossible to verify a specific mixing ratio by conducting full LLM pre-training. A more feasible approach is to adjust the data mixing ratios on the fly, *i.e.,* starting from an LLM360 checkpoint, and either increasing or decreasing a specific data ratio from a particular category, *e.g.,* increasing the data weight in Wikipedia.

- Studies of trustworthiness in LLMs can benefit substantially from access to intermediate checkpoints, plus controllability of trustworthiness (Qian et al., 2024).

- For building domain-specific LLMs (*e.g.,* medical, finance, law, etc.), one may not necessarily want to start from the final LLM checkpoint (which is more akin to fine-tuning). Instead, one can always pick one of the LLM360 checkpoints (*e.g.,* from 50% of the pre-training stage) and resume the pre-training to obtain a domain-specific LLM.

- Many algorithmic approximation frameworks for efficient training require partially trained model weights (Wang et al., 2021; 2023). LLM360 can provide suitable model initializations for these methods.

**Issues Encountered During Pre-training.** We believe that it is important to be fully transparent and discuss issues encountered during pre-training, which can be a major help to the community. We discuss several major issues encountered during the pre-training process of AMBER in Appendix § E. These issues could potentially impact our final model performance, and we aim to address most of these issues in subsequent LLM pre-training efforts.

**Carbon Footprint.** For AMBER, the estimated GPU-only power consumption is 63.9 MWh, representing a carbon footprint of 24.6 tCO2eq. We do not have the specifications to estimate the full system consumption. Estimating chip-only consumption for Cerebras CS-2 systems is difficult, but the estimated total carbon emission for CRYSTAL is at a similar scale of AMBER.

**LLM360 and Responsible Usage.** Given the wide-ranging applicability and high performance of LLMs, it is essential to carefully manage the potential impact and risks associated with them. We believe the transparent nature of LLM360 can help make the potential risks known to all stakeholders. As one example, one form of risk associated with LLMs includes biases (Weidinger et al., 2022), such as social stereotypes, discrimination and exclusion, that may result from under-representation of certain languages or domains. By inspecting the exact training data and bias analysis (e.g., BOLD (Dhamala et al., 2021)) in ANALYSIS360 (also see Appendix A), stakeholders can have a thorough review of these risks before deploying the models. LLM360 can also help with risk mitigation. The project shares reproducible traces and exact data during LLM training, providing a reusable environment for researchers to conduct experiments to design better guardrails to contain potential risks.

## 7 Conclusion and Future Work

In this paper, we introduced LLM360, an initiative for comprehensive and fully-transparent open-source large language models. As an initial effort, we have released two pre-trained 7B LLMs: AMBER (a general-purpose English language LLM) and CRYSTAL (an LLM pre-trained with a focus on code generation), along with the fine-tuned models AMBERCHAT, AMBERSAFE, and CRYSTALCHAT. In terms of artifacts, we released pre-training code, configurations, hyperparameters, intermediate model checkpoints, optimizer states, as well as the data sequence and data processing code. Our vision is to significantly advance and promote transparency within the open-source LLM pre-training community.

For future work, we are conducting a more detailed analysis of AMBER and CRYSTAL's base models as well as their fine-tuned models. Detailed results will be released and discussed in their respective technical reports. Our team is also pre-training a much larger LLM, which will be fully released as soon as the pre-training is complete. Additionally, we will explore the optimal ratios for mixing different subsets in the pre-training datasets.

## Acknowledgement

We would like to thank Natalia Vassilieva, Joel Hestness, William Marshall, and Bhargav Kanakiya for their contribution to CRYSTAL and support on the LLM360 project, the MBZUAI team for providing and managing the computing infrastructure, and anonymous reviewers.

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

# Appendix

## A    Responsible Research and Ethics Statement

LLM360 is created with the mission to train and release open-source large language models to foster transparency, trust, and collaborative research. While large language models have demonstrated promise in advancing numerous domains throughout commercial and academic settings, the technology is still relatively poorly understood. Due to the significant capital requirements to training and experimentation with LLMs, many learnings in the space happen behind closed doors. The lack of knowledge transfer will have negative effects for the ecosystem as advances will be limited to small groups. To fully realize the potential large language models can deliver, we believe that the core tenets of transparency, trust, and collaboration are paramount to the long term success of the field.

For each model released under LLM360, we release the datasets, data preparation scripts, training code, numerous intermediate checkpoints, and complete analysis. We prioritize publicly available datasets such as The RedPajama (Together Computer, 2023b) and Refined Web (Penedo et al., 2023) and existing architectures and conventions such as LLaMA (Touvron et al., 2023a) to make our resource relevant and easy to access. By providing the listed artifacts, we hope to promote the reproducibility for all our work to encourage additional research.

Datasets are expensive to curate and are a major competitive advantage for training performant models. By making all data available, our models are fully auditable. We provide clarity on all pretraining sources, the ethical manner in which data was sourced, and the actual data. Releasing checkpoints from the entire training process enables fine grained research into training dynamics (Qian et al., 2024) which would otherwise be restricted to those with the financial resources to pretrain models. We believe that the future should only be constrained by our creativity, not man-made hurdles, and hope that access to our artifacts motivates others to pursue their own creative research unhindered.

**Ethical Use.**    We release our scores for LLM360 models on safety and bias evaluations such as Toxigen (Hartvigsen et al., 2022), BOLD (Dhamala et al., 2021), and TruthfulQA (Lin et al., 2022). These scores educate users on the potential risks that using generated text from our models. We present the results in Table 5. Our safety-tuned model, AMBERSAFE, shows a significant reduction in toxicity, as evidenced by its lower toxicity score on the Toxigen dataset. Both models also achieve high scores on the BOLD dataset, indicating that they generate higher sentiment values across various domains, including race, gender, religion, political ideology, and profession. Additionally, the models exhibit a low standard deviation in the race category, suggesting that they do not exhibit strong biases across different races.

Additionally, we gather our data from reputable sources and apply standard filtering to remove harmful data, but we cannot guarantee the outputs of our models will be completely safe. All users should conduct their own testing before adopting our models. LLM360 has released our processed versions (involving mixing and shuffling) of publicly available datasets such as C4, RedPajama, and SlimPajama. As such, we have not introduced any new data that could cause additional ethical concerns such as leaking personal identification information. We are committed to continuously monitoring the policies and status of these public datasets, ensuring full compliance with applicable laws and guidelines.

| The LLMs | Toxigen | BOLD Avg. | BOLD Race Std. |
|---|---|---|---|
| AMBERSAFE (7B) | 2.0 | 0.51 | 0.092 |
| AMBERCHAT (7B) | 10.77 | 0.64 | 0.069 |
| Llama2-7B-base | 21.28 | 0.304 | - |
| Llama2-7B-chat | 0.01 | 0.482 | - |
| Code Llama 7B | 22.64 | 0.230 | - |
| Code Llama 7B Instruct | 0.04 | 0.503 | 0.042 |
| Falcon-7B-base | 14.53 | 0.283 | - |
| Falcon-7B-instruct | 5.78 | 0.332 | 0.035 |
| MPT-7B-base | 22.32 | 0.32 | - |
| MPT-7B-instruct | 16.33 | 0.302 | - |

Table 5: A comparison on safety/bias, showing AMBERCHAT, AMBERSAFE, and a few reference models.

LLM360 models are also trained with coding abilities. When using code generated from large language models, users should always review the output before submitting it into their codebase. Generated code may introduce issues such as insecure code which cannot be eliminated from the model. Users should perform their own safety testing and code reviews before deploying applications.

## B  Model Architecture and Hyperparameter Details

| Model Details | AMBER | CRYSTAL |
|---|---|---|
| Number of Parameters | 6.7B | 6.7B |
| Hidden Size | 4096 | 4096 |
| Intermediate Size (in MLPs) | 11008 | 10922 |
| Number of Attention Heads | 32 | 32 |
| Number of Hidden Layers | 32 | 32 |
| LR Schedule | Cosine Decay | Linear Decay |
| Normalization | RMSNorm | LayerNorm |
| Activation | SwiGLU | SwiGLU |
| Sequence Length | 2048 | 2048 |
| Vocabulary Size | 32000 | 32032 |
| Position Embedding | Rotary | Rotary (25%) |
| Bias | None | Linear & LayerNorm |
| QK Dot Product Scaling | $\frac{QK^T}{\sqrt{d}}$ | $\frac{QK^T}{d}$ |
| Model Parametrization | Standard | $\mu P$ (Yang et al., 2022) |
| Warmup Steps | 2000 | multi-phase |
| Batch Size | 2240 | 2112 |

Table 6: Architecture and configuration for AMBER and CRYSTAL models.

**AMBER.**  We choose AMBER pre-training hyperparameters by following LLaMA as close as possible (Touvron et al., 2023a). We use RMSNorm (Zhang & Sennrich, 2019a) with $\epsilon = 1e^{-6}$. We use the AdamW optimizer with $\beta_1 = 0.9$ and $\beta_2 = 0.95$, the initial learning rate is set to $\eta = 3e^{-4}$, following a cosine learning rate schedule that decreases to a final rate of $\eta = 3e^{-5}$. We apply a weight decay of 0.1 and use gradient clipping at 1.0. The model is warmed up over 2,000 steps. Differing from the LLaMA setup, based on our hardware setting with 224 GPUs, we use a pre-training batch size of 2,240 (224 × 10) instead of 2,048. We uses FlashAttention (Dao et al., 2022) for fast attention computation.

**CRYSTAL**  The architecture of CRYSTAL is adapted from prior work such as GPT-2 (Radford et al., 2019), GPT-NeoX (Andonian et al., 2023), LLaMA, and BTLM (Dey et al., 2023), featuring decoder-only models comprising 32 layers.

The model is trained on a non-GPU hardware architecture with Cerebras Condor Galaxy 1 (CG-1), hence some of the model architetures differ from the common choices. Due to the memory layout of the hardware, the model can be trained efficiently without using RMSNorm.

We incorporate a novel enhancement known as maximal update parameterization ($\mu P$), as described by Yang et al. (2022), enabling uniformity in hyperparameters including optimization-related hyper-parameters, *i.e.*, *learning rate, batch size, Adam coefficient*, etc., and initialization-related hyper-parameters across models of varying widths.

The $\mu P$ hyperparamter scalling are allpied to the following parameters.

1. Input embeddings are scaled by `mup_embeddings_scale`.
2. Output logits are scaled by `mup_output_alpha` × `mup_width_scale`.
3. Attention weights scaling is refined to division by the hidden dimension size $\left(\frac{QK^T}{d}\right)$ instead of its square root $\left(\frac{QK^T}{\sqrt{d}}\right)$.
4. Learning rates and weight decay are optimized for different parameter groups:
   - Embedding layer: LR=BASE_LR, WD=BASE_WD.
   - Normalization layers: LR=BASE_LR, WD=0.
   - Other Parameters: LR=BASE_LR × `mup_width_scale`, WD=BASE_WD.
5. Initialization ranges are determined based on $\mu P$ hyperparameters chosen based on a hyperparameter sweep.
   - `mup_initialization_standard_deviation`: 0.073
   - `mup_embeddings_scale`: 14.6
   - `mup_output_alpha`: 2.22
   - `mup_width_scale`: 0.0625

The pretraining of CrystalCoder is conducted in three stages, each with a distinct dataset and learning rate schedule. The pretraining configuration is detailed in Table 7. The warmup steps are heuristically picked to be around 1% of the training steps, except for Phase 2 where we adopt most of the settings from Phase 1.

|  | Phase 1 | Phase 2 | Phase 3 |
|---|---|---|---|
| warmup steps | 86 | 86 | 276 |
| total steps | 79721 | 214387 | 27590 |
| max LR | 0.012 | 0.0087825 | 0.002 |
| min LR | 0.0086628 | 0.00013679 | 0.0002 |
| optimizer | – | AdamW | – |
| beta1 | – | 0.9 | – |
| beta2 | – | 0.95 | – |
| epsilon | – | $1\times10^{-9}$ | – |
| weight decay | – | 0.1 | – |
| gradient clip | – | 1.0 | – |
| batch size | – | 2112 | – |
| sequence length | – | 2048 | – |
| trained tokens (current stage) | 0.345T | 0.927T | 0.1T |
| trained tokens (accumulated) | 0.345T | 1.272T | 1.372T |

Table 7: Pretraining configuration. We choose the warmup steps to be approximately 0.1% of the total steps in Phase 1, except for Phase 2, we reuse the same numbers. In Phase 3, we set it to be 1% of the total steps.

## C  Amber Infrastructure Details

AMBER is trained on a GPU cluster of 56 DGX A100 nodes, each equipped with 4× 80GB A100 GPUs. Each GPU is connected with 4 links NVLink. Cross node connection setting is 2 port 200 Gb/sec (4× HDR) InfiniBand. The throughput we achieved with our distributed training framework is around 582.4$k$ tokens per second. Our pretraining framework is lit-llama[7] developed based on PyTorch Lightning. We used mixed-precision during pre-training with BF16 for activations and gradients and FP32 for model weights (Micikevicius et al., 2017).

---

[7]https://github.com/Lightning-AI/lit-llama

## D   Additional Memorization Analysis

Here we provide additional details about our analysis on memorization over the sequence of LLM360 model checkpoints. As mentioned above, to show an example of the analysis that can be performed over the set of model checkpoints, we conduct an initial study on memorization in LLMs. Recent work (Carlini et al., 2021; 2022) shows that LLMs may memorize a significant part of their training data, which can be extracted with appropriate prompting. Such memorization not only raises privacy concerns in leaking private training data, but also downgrades the performance of LLMs if the training data contains unintended duplicates or peculiarities. As we release all checkpoints and data, we can conduct a comprehensive analysis of memorization across the whole stage of training.

We adopt the *memorization score* introduced in (Biderman et al., 2023a), indicating the accuracy of tokens in the continuation of length $l$ with a prompt of length $k$,

$$\text{score}(k,l) = \frac{1}{l}\sum_{i}^{l}\mathbf{1}[S_{k+i} = G_{k+i}],$$

where $S_{0:k+l}$ is the sequence from training data, while $G_{k:k+l}$ is the generated sequence with prompt $S_{0:k}$. A *memorized* or $k$-extractable (Carlini et al., 2021) sequence has a memorization score of 1. Following (Biderman et al., 2023a;b), we conduct our experiments with $k = l = 32$. We sampled 1000 sequence from each of the 360 data chunks, and use the first 64 tokens of each sequence to conduct the following experiments.

We show the distribution of memorization scores for 10 selected checkpoints in Figure 6 (left), and additionally annotate the percentage of score $= 1$. For every checkpoint, we only include the data chunks it has already been trained on. From the result, we learn that 1) More than 1% of the sequences are 32-extractible from AMBER; 2) AMBER can memorize more sequences as training progresses; 3) The spike at score $= 1$ indicates that AMBER can memorize a much larger number of tokens than our preset threshold 32 (consistent with prior work (Carlini et al., 2022; Biderman et al., 2023a)).

We group the data chunks according to the selected checkpoints, and plot the memorization score on each data chunk group for each checkpoint in Figure 6 (right). We find that 1) AMBER checkpoints memorize the latest seen data much more than previous data; 2) For each data chunk, the memorization score drops a bit with additional training, but keeps increasing afterwards.

In addition, we show the correlation between sequences in terms of memorization score or $k$-extractable metric in Figure 7. We witness a strong correlation between the checkpoints.

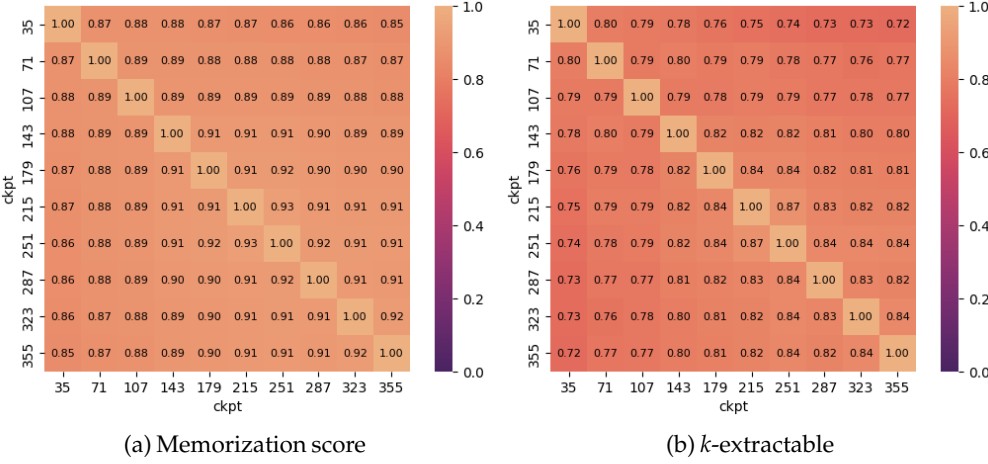

(a) Memorization score                              (b) $k$-extractable

Figure 7: The correlation of sequences in terms of memorization score and $k$-extractable metric between each of the checkpoints.

# E    Issues Encountered During Training

In this section, we discuss several major issues encountered during the pre-training process of AMBER. These issues could potentially impact our final model performance. We have addressed most of these issues in subsequent LLM pre-training efforts.

**NaN loss on a few data chunks.**    During the pre-training procedure, we encountered NaN loss in four out of 360 data chunks. Whenever we faced this issue, we tentatively skipped the entire data chunk. Initially our plan was to train on these four data chunks in later stage of the training, however, we found that these data chunks tend to cause NaN loss regardless of the position of training. We end up finishing our training by taking the first four chunks from the training sequence to complete our learning rate schedule.

**Missing optimizer states.**    In our pre-training framework, we did not manage to save the optimizer states; we only saved model checkpoints for each data chunk. This oversight might be the cause of the NaN loss issue observed in the four data chunks, as mentioned earlier. Each time we resumed pre-training from a previous model checkpoint, the optimizer state in the AdamW optimizer was re-initialized. This re-initialization could potentially affect model training stability.

**Discrepancies on the precision of checkpoints.**    In the initial phase of pre-training, our codebase had an issue where model checkpoints were saved with BF16 precision, despite our mixed precision training process maintaining model weights at FP32. This issue was later identified and rectified by our team, ensuring that all subsequent model checkpoints were saved with FP32 precision. We anticipate that the initial BF16 model checkpoints may have contributed to some degree of accuracy drop in the model.

# F    Additional comparisons against other LLMs

We also provide additional evaluations of our models against other open-weight and open-source language models. In Table 8, we evaluate all models using the exact OpenLLM leaderboard settings.

In comparison, AMBER performs similar to open source models released around the same time (Falcon, MPT, Incite) with a slight advantage on MMLU but weaker in other metrics like ARC. CRYSTAL, our newer model, is comparable with other models released around the same time (Llama2, OpenLlama), where it shows a strong MMLU score of 48.78, as compared to Llama2's 43.80. As LLMs are advancing quickly, newer models in the list, such as Gemma, Qwen1.5 and Olmo 1.7 are generally better.

To study the effect of instruction tuning and safety alignment, we compute the MT-bench (Zheng et al., 2023) score of AMBER, AMBERSAFE and AMBERCHAT, as long as a few reference models.

To evaluate our models on further metrics, such as coding, we have conducted additional evaluations with newer version of LM-Harness and other benchmark suite (the exact evaluation settings are available together with our released evaluation code). We show evaluation results for these metrics in Table 10.

# G    Finetuning Details

## G.1    Prompt Format

**AMBER prompt format.**    We use the Vicuna conversation template from FastChat.

```
USER: first user utterance ASSISTANT: first model response 
USER: next user utterance ASSISTANT: next model response 
```

Note that  is the EOS symbol for AMBER

| Model | ARC-C | HellaSwag | MMLU | TruthfulQA |
|---|---|---|---|---|
| AMBER | 41.89 | 71.63 | 30.76 | 34.00 |
| CRYSTAL | 47.01 | 71.97 | 48.78 | 35.91 |
| Llama1-7B | 50.94 | 77.80 | 35.67 | 34.34 |
| Llama2-7B | 53.07 | 77.74 | 43.80 | 38.98 |
| Mistral-7B | 59.98 | 83.31 | 64.16 | 42.15 |
| Gemma-7B | 61.09 | 82.20 | 64.56 | 44.79 |
| Qwen1.5-7B | 54.18 | 78.51 | 61.97 | 51.08 |
| Codellama-7B | 39.93 | 60.80 | 31.12 | 37.82 |
| OpenLlama-v2-7B | 43.69 | 72.20 | 41.29 | 35.54 |
| Olmo-7B | 45.65 | 77.31 | 28.13 | 35.93 |
| Olmo-1.7-7B | 49.40 | 78.68 | 53.52 | 35.89 |
| Falcon-7B | 47.87 | 78.13 | 27.79 | 34.36 |
| MPT-7B | 47.70 | 77.57 | 30.80 | 33.44 |
| RedPajama-Incite-7B | 46.25 | 71.63 | 27.68 | 33.03 |

Table 8: A comparison of various models across OpenLLM leaderboard metrics: ARC-C, HellaSwag, MMLU, and TruthfulQA.

| Model | MT-Bench Score |
|---|---|
| AMBER | 2.48750 |
| AMBERCHAT | 4.956250 |
| AMBERSAFE | 4.971264 |
| Mistral-7b-sharegpt-90k | 6.787975 |

Table 9: MT-Bench Scores for Amber model family and reference models, averaged over 3 runs. We compared it with Mistral 7B trained on ShareGPT-90K, to get a rough perspective.

**CRYSTAL prompt format** We introduced four special tokens to the tokenizer. These tokens were integrated into the existing 32032-token vocabulary without an expansion, leveraging reserved vocabulary slots. The conversation is wrapped by the tokens  and , framing the structure as follows:

```
 <|sys_start|> system prompt <|sys_end|> <|im_start|> first user utterance <|
    ↪ im_end|> first model response <|im_start|> next user utterance <|im_end|>
    ↪ next model response 
```

## G.2 CRYSTALCHAT Finetuning Datasets

Table 11 summarizes the CRYSTALCHAT dataset collection.

| Model | Openbook QA | RACE | BoolQ | PIQA | HumanEval p@1 | MBPP p@1 | Winogrande | GSM8K |
|---|---|---|---|---|---|---|---|---|
| AMBER | 40 | 37.70 | 68.90 | 79.43 | - | - | 64.25 | - |
| CRYSTAL | 41.20 | 38.18 | 74.43 | 78.07 | 23.90 | 30.99 | 67.01 | 12.36 |
| Llama2-7B | 44.20 | 39.52 | 78.07 | 78.78 | 13.05 | 20.09 | 69.38 | 14.71 |
| Llama1-7B | 44.40 | 40.28 | 75.01 | 78.94 | 10.61 | 17.04 | 70.24 | 8.87 |
| CodeLlama-7B | 36.80 | 39.52 | 74.65 | 72.58 | 30.06 | 39.20 | 65.51 | 11.15 |
| Mistral-7B | 44.20 | 40.86 | 83.73 | 82.15 | 29.11 | 38.78 | 74.19 | 37.68 |
| Falcon-7B | 43.80 | 37.42 | 73.70 | 80.57 | 9.42 | 13.38 | 67.24 | 4.62 |
| MPT-7B | 42.00 | 38.66 | 74.00 | 80.30 | 16.52 | 22.49 | 68.51 | - |
| Olmo-7B | 42.60 | 38.37 | 72.66 | 79.92 | 14.02 | 14.40 | 68.90 | 4.09 |
| Pythia-6.9B | 25.50 | - | 62.10 | 75.20 | 7.68 | 6.00 | - | - |
| Starcoder-13B | 32.80 | 32.15 | 63.91 | 65.72 | 30.70 | 37.56 | 54.30 | 9.02 |
| Phi1.5 | 48.20 | 37.51 | 74.74 | 75.95 | 35.36 | 35.19 | 72.92 | 31.23 |

Table 10: A comparison of various models across additional evaluations: Openbook QA, RACE, BoolQ, PIQA, HumanEval p@1, MBPP p@1, Winogrande, and GSM8K.

| Subset | #Tokens | Avg. #Q | Avg. Q Len | Avg. #R | Avg. R Len |
|---|---|---|---|---|---|
| OASST1-guanaco | 4,464,640 | 1.36 | 38.28 | 1.36 | 271.69 |
| SlimOrca | 225,628,160 | 1.00 | 259.16 | 1.00 | 151.12 |
| ShareGPT | 112,914,432 | 3.28 | 94.53 | 3.64 | 365.81 |
| Evol-ShareGPT | 85,954,560 | 1.00 | 145.99 | 1.00 | 425.17 |
| ChatLogs | 29,337,600 | 3.39 | 95.58 | 3.24 | 191.42 |
| CodeAlpaca | 2,623,488 | 1.00 | 32.46 | 1.00 | 67.68 |
| Rosetta Code | 7,987,200 | 1.00 | 450.09 | 1.00 | 533.52 |
| Evol-CodeAlpaca 1 | 73,803,776 | 1.00 | 210.33 | 1.00 | 437.92 |
| Evol-CodeAlpaca 2 | 34,910,208 | 1.00 | 114.99 | 1.00 | 300.29 |
| WebAlpaca | 43,673,600 | 1.00 | 96.29 | 1.00 | 746.52 |
| General Textbooks | 85,590,016 | Not instruction data | | | |
| Programming Books | 395,628,544 | Not instruction data | | | |
| Total | 1,102,516,224 | | | | |

Table 11: Dataset Statistics. Q stands for Query. R Stands for reply.

