# OpenReview forum: "LLM360: Towards Fully Transparent Open-Source LLMs"
_colmweb.org/COLM/2024/Conference — COLM_

### Official Review · Reviewer_13mu · 2024-05-09

**Rating:** 8
**Confidence:** 3
**Ethics Flag:** 1

**Summary:**

This paper presents a fully open-sourced language modeling project, whose aim is to release all code, training data, intermediate checkpoints and results, and also serves as a technical report for two 7B parameter language models, Amber (trained on 1.3T tokens of English text) and Crystal (trained on 1.5T tokens of English text and code). Along with these pretrained models, three finetuned variants of these models, AmberChat and CrystalChat (SFT) and AmberSafe (DPO) were described and evaluated. Amber and AmberChat were evaluated on ARC, Hellaswag, MMLU, and TruthfulQA, and Crystal was evaluated additionally on HumanEval (from the Codex paper) and MBPP.

**Reasons To Accept:**

The paper presents a strong motivation for fully open language model research, and includes many useful training details. Moreover trained models and the intermediate checkpoints and WandB logs are useful artifacts for the community.

**Reasons To Reject:**

The paper could use more detailed evaluations and further discussion of existing results. CrystalChat does not have any evaluations in the paper currently.

1. Chat-focused evaluation (e.g. AlpacaEval) and safety evaluations (e.g. Toxigen) of AmberChat, CrystalChat and AmberSafe would helpful to see.
2. Further discussion of the performance on Amber (the base model) on the OpenLLM leaderboard benchmarks would be informative. What would make the model's performance more comparable to better 7B models? Is it the training data mix or the size? I understand that concrete observations cannot be made here because all the training details of the other models are not known, but a discussion on how the model could be improved would be helpful.
3. The draft currently does not have any details of the carbon footprint of training the models.

---

> ### Author Rebuttal · Authors · 2024-05-31
>
> Thank you for the thoughtful review of our paper, and for the helpful feedback. We respond back to individual comments below.
>
> ### **CrystalChat evaluations**
> Thanks for pointing this out (this was an accidental omission). We have a full suite of evaluations for CrystalChat, which we will include in our paper, including the Open LLM Leaderboard metrics suite (e.g. ARC-C, GSM8K, HellaSwag, MMLU, TruthfulQA, Winogrande) and coding evaluations (HumanEval, MBPP). For reference, CrystalChat performs on par with Llama-2-7b-Chat on the language evaluations, and on par with CodeLlama-7b-Instruct for the code evaluations—e.g. for code evaluations, CrystalChat scores: HumanEval (Pass@1): 34.12, & MBPP (Pass@1): 39.11.
>
> ### **Chat-focused evaluations**
> We also have completed chat-focused evaluations, specifically the MT-Bench evaluation. For reference, for a 3-round average on MT-Bench, Amber scores 2.49, AmberChat scores 4.95, and AmberSafe scores 4.97. We will describe this evaluation and include all results in our paper.
>
> ### **Safety evaluations**
> We have conducted an evaluation on safety and toxicity of our model, which we will add to our paper. This analysis includes evaluations on BOLD [1] (AmberChat: 0.51, AmberSafe: 0.64, Llama2: 0.46; higher is better sentiment) & ToxiGen [2] (AmberChat: 10.8, AmberSafe: 2.0, Llama2: 21.28; higher is more toxicity).
>
> [1] https://arxiv.org/abs/2101.11718
> [2] https://arxiv.org/abs/2203.09509
>
> ### **Discussion of performance metrics**
> We have collected some evidence explaining Amber’s performance, and ways to improve it. One important aspect is simply the data. Although our compute budget did not allow for extensive ablations on data composition, we found (in subsequent training, such as in Crystal) that modifying the data composition, size, and quality, produced a significant impact on Open LLM Leaderboard metrics. Another potential explanation is due to a precision issue during training. As described in appendix E, in an initial phase of our training, we used a suboptimal model precision for some of our checkpoints; we believe that this may have also led to some drop in accuracy. We will add these discussions to our paper.
>
> ### **Carbon footprint of models**
> Thank you for the idea! We will complete an estimate of the carbon footprint of our models, and add these to the paper. For example, for our Amber model, we estimate that the power consumption is 63.9 MWh, and our carbon footprint is 24.6 tCO2eq.

---

> > ### Comment · Reviewer_13mu · 2024-06-04
> >
> > I thank the authors for their informative response, and am glad to see the additional evaluations and carbon footprint details. Including these details in the paper would make it significantly stronger. I have increased my score accordingly.

---

### Official Review · Reviewer_WFbL · 2024-05-11

**Rating:** 9
**Confidence:** 4
**Ethics Flag:** 1

**Summary:**

This work present a fully open-source LLMs initiative LLM360, where not just model weights, but all the training data, training code, model checkpoints and intermediate results are made available to the community. This initiative is significant for the community as it addresses several important challenges LLM research is dealing with - data provenance, reproducibility and open collaboration.

In the paper, the authors outline the LLM360 approach, as well as pretrain two new LLMs (Amber and Crystal) from scratch and release them under the LLM360 framework, together with all the associated core open-sourced artifacts. The paper is very well written and clearly structured, the rationale behind open-sourcing the core artifacts is properly explained, the process of pre-training newly presented LLMs and their evaluation are both discussed in sufficient details. The take-home messages and potential use cases of LLM360 make this work useful also for the future researchers and practitioners. The authors are transparent about the issues encountered during training and make suggestions about the responsible usage of their models.

**Questions To Authors:**

Correct the typo under the Issues Encountered During Pre-training section on page 9: "...which can be a major help ***the*** the community".

**Reasons To Accept:**

- this ambitious work contributes to a more inclusive and collaborative research environment by reinforcing visibility and access to LLM training, fine-tuning, and evaluation processes.
- this work has a strong technological impact, as it presents two newly pre-trained from scratch open-source LLMs with the associated open training datasets, data processing code, training code, hyperparameters, configurations, model checkpoints, collected logs and metrics. These high quality resources will enable future high quality and impactful work.
- the described pre-training process of two new LLMs presents a strong empirical foundation. It provides useful insights for the community and is beneficial for understanding the details and potential major issues of the pre-training process.

**Reasons To Reject:**

I do not see significant reasons to reject this paper. In order to improve it for the camera-ready version, I would suggest adding a couple of recent models to the related work discussion, specifically Aya (Üstün, Ahmet, et al. Aya model: An instruction finetuned open-access multilingual language model", 2024) and StarCoder 2 (Lozhkov, Anton, et al. "StarCoder 2 and The Stack v2: The Next Generation", 2024 ).

It would also be beneficial to read a bit about how the authors see the future of LLM360 in terms of the open collaboration with the community, e.g. do they expect external contributions to LLM360 in the future?

---

> ### Author Rebuttal · Authors · 2024-05-31
>
> Thank you for the thoughtful review of our paper, and for the helpful feedback. We respond back to individual comments below.
>
> ### **Additions to related work**
> Thanks for the additional references to recent models for the related work discussion. We have updated our paper to include these new citations.
>
> ### **Future of LLM360 and open collaboration**
> We are indeed very enthusiastic about expanding the breadth of the LLM360 project, and have been actively working to increase the number of collaborators (and collaborating organizations) in this project. As a fully-open community, we welcome anyone interested to join our collaboration, and are working on processes and governance that can scale up to greater numbers of members.
>
> ### **Typo**
> Thank you as well for pointing out this typo. We have fixed this in our paper!

---

> > ### Comment · Reviewer_WFbL · 2024-06-05
> >
> > I thank the authors for the response. I am still confident with my score and would be glad to see this paper published.

---

### Official Review · Reviewer_wBF5 · 2024-05-11

**Rating:** 6
**Confidence:** 5
**Ethics Flag:** 1

**Summary:**

The LLM360 initiative addresses a critical need in the AI research community for complete transparency in the training of Large Language Models (LLMs). The quality of this work is evident in the comprehensive approach to open-sourcing, which includes training code, data, model checkpoints, and intermediate results.

**Questions To Authors:**

1. How do you plan to sustain the LLM360 initiative in terms of funding and resource allocation?
2. Can you elaborate on any legal or ethical considerations in open-sourcing such extensive datasets and model details?

**Reasons To Accept:**

1. LLM360’s full disclosure of training processes, including intermediate checkpoints and data provenance, addresses a significant gap in current LLM research practices.
2. By providing all necessary artifacts for replication, LLM360 significantly enhances the reproducibility of LLM research, aligning with scientific principles.

**Reasons To Reject:**

1. The paper lacks extensive comparative analysis with existing LLMs, which could better demonstrate the efficacy and efficiency of the LLM360 models.
2. The long-term sustainability of continually open-sourcing such extensive resources is not discussed, which could be a potential hurdle in widespread adoption.

---

> ### Author Rebuttal · Authors · 2024-05-31
>
> Thank you for the thoughtful review of our paper, and for the helpful feedback. We respond back to individual comments below.
>
> ### **Comparative analysis with existing LLMs**
> We have conducted broader evaluations of Amber and Crystal, and we will add these results to the final paper, along with a comparative analysis with results reported by other existing LLMs. Specifically, these evaluations include the Open LLM Leaderboard metrics suite (e.g. ARC-C, GSM8K, HellaSwag, MMLU, Winogrande, etc), additional evaluations (BoolQ, DROP, Openbook QA, PIQA), HumanEval metrics, and MBPP metrics.
>
> ### **Discussion of long-term sustainability**
> We also agree on the importance of sustainability of this project. In the near-term, we are engaging a broader community beyond the LLM360 team, and have secured resources for several newer models. In the longer term, our vision for LLM360 is to promote transparency and reproducibility of LLMs to be adopted by many others. We have seen early signs of such adoption by other projects that used our model checkpoints, data, and evaluation framework, and we hope that continued progress on these fronts will lead to longer term continuation of open-source AI in general. We also see other recent efforts that are investing resources for fully open-source LLM training, such as OLMo, and MAP-Neo, so we are optimistic about the community’s collective interests in sustaining these efforts.
>
> ### **Question about legal/ethical considerations**
> We respect the risks associated with releasing open-source models; however, we believe that the final models that we release do not add additional major risks, especially since there exist open-weight models (such as Llama 3) which offer stronger capabilities but at a lower level of transparency. We hope that open-source releases of LLMs will enable researchers to study the security and safety issues associated with models of this scale.
>
> As for legality, we aim to follow all best-practices for licensing of our model. Further, we only leverage existing open-source data, and simply release the exact data sequence (associated with model checkpoints) and our data processing procedures. Indeed, we feel that being fully-transparent about the data used in an LLM is the best way to ensure that an organization is adhering to legal practices.
>
> ### **Thanks**
>
> We hope that we were able to address each of your comments. Please let us know if there is anything else we could answer which would improve your score.

---

> > ### Comment · Reviewer_wBF5 · 2024-06-05
> >
> > I appreciate the author's response. I still have concerns regarding comparative analysis with existing LLMs. It is helpful if the authors could report the results in the response.

---

> > > ### Author Response · Authors · 2024-06-05
> > >
> > > We understand your concern of comparing the model with existing LLMs and agree that a comprehensive report will be helpful. Note that in our paper we compare models with similar conditions (e.g. dataset, release dates), and report the numbers following the exact settings of public OpenLLM leaderboard as possible. We have more evaluation results to share here:
> > >
> > > **Metrics using Open LLM leaderboard setting**
> > >
> > > We have evaluated both models using the exact OpenLLM leaderboard settings as below, the numbers of several other models are referenced from the leaderboard.
> > >
> > > | Model  | ARC-C | HellaSwag | MMLU | TruthfulQA|
> > > | -------- | ------- | ------- | ------- | ------- |
> > > | Amber  | 41.89   | 71.63   | 30.76   | 34.00   |
> > > | Crystal |  47.01   | 71.97   | 48.78   | 35.91   |
> > > | Llama1-7B | 50.94 | 77.80 | 35.67 | 34.34 |
> > > | Llama2-7B | 53.07 | 77.74 | 43.80 | 38.98 |
> > > | Mistral-7B | 59.98 | 83.31 | 64.16 | 42.15 |
> > > | Gemma-7B | 61.09 | 82.20 | 64.56 | 44.79 |
> > > | Qwen1.5-7B | 54.18 | 78.51 | 61.97 | 51.08 |
> > > | Codellama-7B | 39.93 | 60.80 | 31.12 | 37.82 |
> > > | OpenLlama-v2-7B | 43.69 | 72.20 | 41.29 | 35.54 |
> > > | Olmo-7B | 45.65 | 77.31 | 28.13 | 35.93 |
> > > | Olmo-1.7-7B  | 49.4 | 78.68 | 53.52 | 35.89 |
> > > | Falcon-7B | 47.87 | 78.13 | 27.79 | 34.36 |
> > > | MPT-7B | 47.70 | 77.57 | 30.80 | 33.44 |
> > > | RedPajama-Incite-7B | 46.25 | 71.63 | 27.68 | 33.03|
> > >
> > > In comparison, Amber performs similar to open source models released around the same time (Falcon, MPT, Incite) with a slight advantage on MMLU but weaker in other metrics like ARC.
> > >
> > > CrystalCoder, our newer model, is comparable with other models released around the same time (llama2, OpenLlama), where it shows a strong MMLU score of 48.78, as compared to Llama2’s 43.80.
> > >
> > > This field is advancing quickly, and newer models in the list, such as Gemma, Qwen1.5 and Olmo 1.7 are generally better.
> > >
> > >
> > > **Other Metrics**
> > >
> > > To evaluate our models on metrics on additional benchmarks, such as coding, we have conducted additional evaluations with newer version of LM-Harness and other benchmark suite (the exact evaluation settings will be available together with the evaluation code)
> > >
> > > | Model  | Openbook QA  | RACE | BoolQ | PIQA | HumanEval p@1 | MBPP p@1  | Winogrande | GSM8K |
> > > | -------- | ------- | ------- | ------- | ------- | ------- | ------- | ------- | ------- |
> > > | Amber | 40 | 37.70 | 68.90 | 79.43 | - | - | 64.25 | - |
> > > | Crystal | 41.20 | 38.18 | 74.43 | 78.07 | 23.90 | 30.99 | 67.01 | 12.36 |
> > > | Llama2-7B | 44.20 | 39.52 | 78.07 | 78.78 | 13.05 | 20.09 | 69.38 | 14.71 |
> > > | Llama1-7B | 44.40 | 40.28 | 75.01 | 78.94 | 10.61 | 17.04 | 70.24 | 8.87 |
> > > | CodeLlama-7B | 36.80 | 39.52 | 74.65 | 72.58 |  30.06 |  39.20 | 65.51 | 11.15 |
> > > | Mistral-7B | 44.20 | 40.86 | 83.73 | 82.15 | 29.11 | 38.78 | 74.19 | 37.68 |
> > > | Falcon-7B | 43.80 | 37.42 | 73.70 | 80.57 | 9.42 | 13.38 | 67.24 | 4.62 |
> > > | MPT-7B | 42 | 38.66 | 74.00 | 80.30 | 16.52 | 22.49 | 68.51 | - |
> > > | Olmo-7B | 42.60 | 38.37 | 72.66 | 79.92 | 14.02 | 14.40 | 68.90 | 4.09 |
> > > | Pythia-6.9B | 25.5 | - | 62.10 | 75.20 | 7.68 | 6.00 | - | - |
> > > | Starcoder-13B | 32.80 | 32.15 | 63.91 | 65.72 | 30.70 | 37.56 | 54.30 | 9.02 |
> > > | Phi1.5 | 48.20 | 37.51 | 74.74 | 75.95 | 35.36 | 35.19 | 72.92 | 31.23 |
> > >
> > >
> > >
> > > **Safety related metrics**
> > >
> > > For instruct-tuned/chat models, the finetuned data can affect the model performance on tasks significantly. Here we provide a comparison on safety/biase of AmberChat, AmberSafe and a few reference models.
> > >
> > > | Model | Toxigen | BOLD Avg. | BOLD Race std. |
> > > | -------- | ------- |  ------- | ------- |
> > > | AmberSafe | 2.0 | 0.51 | 0.092 |
> > > | AmberChat | 10.77 | 0.64 | 0.069 |
> > > | Llama2-7B-Base | 21.28 | 0.304 | - |
> > > | Llama2-7B-Chat | 0.01 | 0.482 | - |
> > > | Code Llama 7B | 22.64 | 0.230  | - |
> > > | Code Llama 7B Instruct | 0.04 | 0.503 | 0.042 |
> > > | Falcon-7B-Base | 14.53 | 0.283 |  - |
> > > | Falcon-7B-Instruct | 5.78 | 0.332 | 0.035 |
> > > | MPT-7B-Base | 22.32 | 0.32 | - |
> > > | MPT-7B-instruct | 16.33 | 0.302 | - |
> > >
> > > We found that safe tuning with DPO makes Amber less toxic, but not necessarily less biased. Specifically, the BOLD score (sentiment towards certain social groups) shows that AmberChat shows a better sentiment score over different social groups, and has a smaller standard deviation across groups (fair).
> > >
> > > Here, we only show the standard deviation across races, we have conducted more analysis and could include them in the revision. Compared with other models, AmberSafe shows good scores against a few open source models released around the same time. However, we can see that the Llama models, after tuning with their internal safety data, show a much safer behavior.
> > >
> > > Due to resource constraints, we prioritized comparing models with comparable settings and models released around the same time of our models, and avoided comparing with moving targets such as commercial APIs. Please let us know if there are specific results you would like to see.

---

> > > > ### Comment · Reviewer_wBF5 · 2024-06-06
> > > >
> > > > Thank you for the response. Considering the author's response, I have increased my score.

---

### Official Review · Reviewer_XdcN · 2024-05-12

**Rating:** 6
**Confidence:** 4
**Ethics Flag:** 2

**Summary:**

This paper discusses the importance of open-source large language models and makes a step in this direction by proposing LLM360, an initiative which encourages the community to make publicly available training data, code, model checkpoints and intermediate results. The authors make a step in this direction by releasing two 7B LLMs pretrained from scratch for the English language, Amber (1.3T tokens) and Crystal (1.4T tokens), altogether with their pre-training data, code, intermediate checkpoints and analyses. In addition, the authors also release corresponding fine-tuned models AmberChat, AmberSafe, CrystalChat.

**Ethics Concerns Details:**

The authors plan to release open-source models to the community, however the paper does not include a detailed analysis of Discrimination / Bias / Fairness Concerns associated with these models.

**Questions To Authors:**

What guardrails are in place to contain the potential risks associated with the models you plan to release?

**Reasons To Accept:**

The paper makes an important contribution towards transparent open-source LLMs by releasing to the community two English LLM models pre-trained from scratch along with the training data, code, model checkpoints and performance reports. The work is timely and would greatly benefit the open-source research community.

The authors acknowledge issues encountered during pre-training with the goal of being fully transparent about the model training and potential pitfalls.

**Reasons To Reject:**

The paper does not discuss the goal of fine-tuning the models. It is unclear from the paper what AmberChat, AmberSafe, CrystalChat have been finetuned to do and for which tasks they can be readily used.

The authors do not provide an explanation why on some metrics the Amber model does well, while on some other metrics (ARC) the model is behind the curve.

This statement should be clarified: “We also find that our finetuned AMBER models are relatively strong, even compared with other similar models.”

The paper frequently mentions the Pythia (Biderman et al., 2023b) model, but there is no direct comparison between the models of this paper with Pythia.

No discussion related to data quality used for pre-training and safety aspects of the model. The paper would greatly benefit from more detailed discussion of data selection and analysis of model biases, controllability and trustworthiness.

“For future work, we are conducting a more detailed analysis of AMBER and CRYSTAL’s base models as well as their fine-tuned models. Detailed results will be released and discussed in their respective technical reports.” - In my view, this detailed model analysis should not be left to future work, but included in the current paper.

---

> ### Author Rebuttal · Authors · 2024-05-31
>
> Thank you for the thoughtful review of our paper, and for the helpful feedback. We respond back to individual comments below.
>
> ### **Goals of fine-tuning**
> We agree with your comment, and will add in a description of our goals in fine-tuning. In short, for AmberChat and CrystalChat, our goal was to produce an instruction-following model that could serve as a chatbot. For AmberSafe, our goal was to carry out a stage of safety-alignment in addition to instructing-tuning, so that the effects of each stage can be studied. We will discuss full details of the fine-tuning procedure and datasets so that our process and aims are fully clear.
>
> ### **Discussion of performance metrics**
> We have made efforts to understand some of the reasons behind Amber performance, and will include an explanation in our paper. As two examples: we found that improving the data composition and size produced a significant increase in Open LLM Leaderboard metrics where Amber underperformed. Additionally, we found an issue with an incorrect model precision during the initial phase of training Amber (described in Appendix E); once fixed, we found that this also improved model performance on our evaluations.
>
> ### **Comparison with Pythia**
> We have completed a full comparison of our model against Pythia (and other popular LLMs) on over a dozen open benchmarks, and will add this comparison between models to the paper.
>
> ### **Additional analysis**
> We agree with your point about a detailed model analysis in future work, and as you suggest will aim to include the full model analysis within the current paper.
>
> ### **Ethics, and discussion of model safety/bias**
> Thank you for bringing up this important point about ethics considerations. We have conducted an ethics and toxicity evaluation in our model, which we will add to our paper. This analysis includes evaluations on BOLD [1] & ToxiGen [2].  We will also include a discussion on the relevant processes we followed to select training data, and steps we took to assess safety, bias and fairness.
>
> As there exist open-weight models (e.g. Llama 3) which offer stronger capabilities but at a lower level of transparency, we believe that releasing our current models does not add additional major risks to the field. However, in the future, if our trained models become more performant, we do believe that guardrails could be necessary to reduce capabilities before release.
>
> [1] https://arxiv.org/abs/2101.11718
>
> [2] https://arxiv.org/abs/2203.09509

---

### Decision · Program_Chairs · 2024-07-10

**Decision:**

Accept

**Comment:**

The authors introduce LLM360, an effort towards building LLMs entirely in the open. In addition to releasing the code, data, and checkpoints from the training of two 7B models, Amber and Cystal, the authors introduce the LLM360 initiative, which is an ongoing LLM-in-the-open project.

The main weaknesses the reviewers raised include:

- There are already similar open LLM projects, e.g., Pythia, that the authors added a comparison two in rebuttal. Notably, two other more-open-efforts are discussed: OLMo (early 2024) and MAP-Neo (released after COLM deadline).
- A few missing comparisons, e.g., chat benchmarks, carbon footprint, etc. (which the authors added in rebuttal)

Overall, I concur with the reviewer consensus that this work is a strong contribution: I see no reason why another promising open LLM effort should not be presented at COLM, esp. given that this work is mostly-concurrent with OLMo/MAP-Neo (and outperforms the prior similarly open model pythia, which is acknowledged/cited/discussed; the authors also promised quantitative comparisons in the camera ready and this seems feasible.)

[ethics comments from PCs] The authors discuss about bias being a dimension of Analysis360, but do not evaluate on any datasets. Moreover, there is no discussion of Personally Identifiable Information (PII) in training datasets and ways to avoid them while releasing these datasets. It would be useful to either provide quantitive numbers on bias or provide an ethics statement that discusses both bias and PII protection. **Ethics statement is mandatory for the revision of the paper.**